# Rice Genome Resequencing Reveals a Major Quantitative Trait Locus for Resistance to Bakanae Disease Caused by *Fusarium fujikuroi*

**DOI:** 10.3390/ijms20102598

**Published:** 2019-05-27

**Authors:** Do-Yu Kang, Kyeong-Seong Cheon, Jun Oh, Hyoja Oh, Song Lim Kim, Nyunhee Kim, Eungyeong Lee, Inchan Choi, Jeongho Baek, Kyung-Hwan Kim, Nam-Jin Chung, Hyeonso Ji

**Affiliations:** 1Department of Agricultural Biotechnology, National Institute of Agricultural Sciences (NIAS), Jeonju 54874, Korea; 636597@naver.com (D.-Y.K.); cjsrudtjd@gmail.com (K.-S.C.); ospreys@nate.com (J.O.); hja-oh@hanmail.net (H.O.); greenksl5405@korea.kr (S.L.K.); knh702@korea.kr (N.K.); wowlek44@korea.kr (E.L.); inchchoi@korea.kr (I.C.); firstleon@korea.kr (J.B.); biopiakim@korea.kr (K.-H.K.); 2Department of Crop Science and Biotechnology, Chonbuk National University, Jeonju 54896, Korea; njchung@jbnu.ac.kr

**Keywords:** bakanae disease, resequencing, genetic map, quantitative trait locus

## Abstract

Bakanae disease (BD), caused by the fungal pathogen *Fusarium fujikuroi*, has become a serious threat in rice-cultivating regions worldwide. In the present study, quantitative trait locus (QTL) mapping was performed using F2 and F3 plants derived after crossing a BD-resistant and a BD-susceptible Korean japonica rice variety, ‘Samgwang’ and ‘Junam’, respectively. Resequencing of ‘Junam’ and ‘Samgwang’ genomes revealed 151,916 DNA polymorphisms between the two varieties. After genotyping 188 F2 plants, we constructed a genetic map comprising 184 markers, including 175 kompetitive allele-specific PCR markers, eight cleaved amplified polymorphic sequence (CAPS) markers, and a derived CAPS (dCAPS) marker. The degree of BD susceptibility of each F2 plant was evaluated on the basis of the mortality rate measured with corresponding F3 progeny seedlings by in vitro screening. Consequently, *qFfR9*, a major QTL, was discovered at 30.1 centimorgan (cM) on chromosome 9 with a logarithm of the odds score of 60.3. For the QTL interval, 95% probability lay within a 7.24–7.56 Mbp interval. In this interval, we found that eight genes exhibited non-synonymous single nucleotide polymorphisms (SNPs) by comparing the ‘Junam’ and ‘Samgwang’ genome sequence data, and are possibly candidate genes for *qFfR9*; therefore, *qFfR9* could be utilized as a valuable resource for breeding BD-resistant rice varieties.

## 1. Introduction

Bakanae disease (BD), caused by the fungal pathogen *Fusarium fujikuroi*, has become a serious global threat in the majority of the rice-cultivating regions, leading to high (3.0–95.4%) losses in yield [1,2,3,4,5]. “Bakanae” means a “foolish rice seedling” in colloquial Japanese and indicates the unusually early elongation of a seedling induced by pathogen-produced gibberellins [6]. *Fusarium fujikuroi* also produces fusaric acid, a phytotoxin that causes stunted growth [6]. BD exhibits diverse symptoms including seedling blight, root rot, crown rot, stunting, foot rot, seedling rot, and grain sterility [3]. Usually symptoms vary depending on the quantity of gibberellins and fusaric acid produced, which vary according to fungal strains and the resistance of rice varieties [6]. The optimum growth temperature for the pathogen ranges between 27 °C and 30 °C, while 35 °C is optimum for infectious disease development [3]. The incidence of BD is predicted to increase with global warming and rising temperatures.

*Fusarium fujikuroi* is a seed-borne pathogen. The ascospores and conidia adhering to the seed act as the primary source of inoculum, whereas the hyphae infect seedlings through the roots and crown [3]. The treatment of seeds using hot water and fungicides such as prochloraz, benomyl, and thiram is widely used for preventing BD [3]. However, the emergence of fungicide-resistant strains has made disease control challenging [7,8,9]. Therefore, the option of developing of BD-resistant rice varieties is critical for disease control.

Several methods have been developed to screen for BD-resistant germplasms. According to Fiyaz et al. [10], a high-throughput screening of such germplasms was possible by inoculating rice seeds in a fungal spore suspension and allowing the growth of seedlings in a greenhouse for 15 days at a day/night temperature and a humidity of 30 °C/25 °C and 60%/80%, respectively. Another seed inoculation screening method used a tissue embedding cassette and a seedling tray [11] to screen rice germplasm and 11 resistant accessions were detected [12]. Another in vitro seedling screening method used inoculated seeds that were planted on solid Murashige and Skoog (MS) medium in a test tube and grown at 28 °C in a growth chamber [13].

Several quantitative trait locus (QTL) mapping studies have been conducted to better understand BD resistance. Yang et al. [14] reported two QTLs for BD resistance on rice chromosomes 1 and 10 using a japonica/indica double haploid population [14]. Three QTLs (*qBK1.1*, *qBK1.2*, and *qBK1.3*) on chromosome 1 were detected using recombinant inbred lines derived from two indica rice parental varieties; namely ‘Pusa 1342’ and ‘Pusa Basmati 1121’, the former being a highly resistant variety and the latter a highly susceptible variety [6]. Hur et al. [15] mapped a major QTL (*qBK1*) in a 520 kb region between *RM8144* and *RM 11295* on chromosome 1 using 168 near-isogenic rice lines (BC6F4) derived from a cross between ‘Shingwang’ and ‘Ilpum’, a highly resistant (indica) and a highly susceptible variety (japonica), respectively. Ji et al. [16] also mapped a major QTL (*qFfR1*) in a 22.56–24.10 Mbp region on chromosome 1 using 180 F2:F3 lines derived from a cross between a resistant Korean japonica variety, ‘Nampyeong’, and a susceptible Korean japonica line, ‘DongjinAD’. A QTL analysis using recombinant inbred lines derived from a cross between the resistant ‘Wonseadaesoo’ variety and the susceptible ‘Junam’ variety revealed a major QTL (*qBK1^WD^*) located at a 1.59 Mbp interval, delimited between chr01_13542347 (13.54 Mbp) and chr01_15132528 (15.13 Mbp), and the pyramiding of *qBK1^WD^* and *qBK1* was performed [17]. In addition, a genome-wide association study using 138 japonica rice accessions revealed two major QTLs, *qBK1_628091* and *qBK4_31750955*, on chromosomes 1 and 4 [18].

Using RNA-seq, it was demonstrated that PR1, germin-like proteins, glycoside hydrolases, mitogen-activated protein kinases, and WRKY transcription factors are upregulated in response to *F. fujikuroi* in the resistant variety Selenio as compared with the susceptible variety ‘Dorella’ [19]. Siciliano et al. [20] reported that *F. fujikuroi* induced an increase in the production of phytoalexins, particularly sakuranetin, in the resistant cultivar ‘Selenio’, while it induced the production of gibberellin and abscisic acid and inhibited the production of jasmonic acid, leading to very low levels of phytoalexins in the susceptible variety ‘Dorella’. A high-quality genome sequence of *F. fujikuroi* revealed gene clusters for the biosynthesis of secondary metabolites such as gibberellin and fumonisin [21]. It has been proposed that gibberellin suppresses the jasmonic acid signaling pathway and promotes infection by necrotrophic pathogens such as *F. fujikuroi* [22,23].

In this study, we aimed to find novel QTLs for resistance to BD which will be used for breeding resistant rice varieties. We performed QTL mapping using F2 and F3 plants derived from crossing ‘Samgwang’ and ‘Junam’, BD-resistant and BD-susceptible Korean japonica rice varieties, respectively. The genotyping of F2 plants was performed using primarily kompetitive allele-specific PCR (KASP) markers developed from single nucleotide polymorphisms (SNPs) detected among Korean japonica rice varieties [24]. A major QTL, *qFfR9*, was discovered on rice chromosome 9, and it may prove useful for breeding BD-resistant rice varieties.

## 2. Results

### 2.1. Phenotype of Parental Varieties and Progenies

In vitro seedling screening was used for investigating the response of ‘Samgwang’ and ‘Junam’ rice varieties to BD. In the case of ‘Junam’, the heights of the treated plants were greater than the control plants until 12 days after inoculation (DAI); subsequently, the heights of plants in both groups were observed to be similar (Figure 1a). In the case of ‘Samgwang’, the heights of the treated plants were greater than the control plants at all time-points during the testing period (Figure 1b). Expectedly, the mortality rates of control ‘Junam’ and ‘Samgwang’ plants remained at 0% throughout the testing period. The mortality rate of treated ‘Junam’ plants sharply increased from 16 DAI and reached 100% at 32 DAI. Conversely, the mortality rate of treated ‘Samgwang’ plants was 0% at up to 32 DAI (Figure 1c). While a majority of the treated ‘Junam’ plants died, all the treated ‘Samgwang’ plants survived at 24 DAI (Figure 1d).

The degree of BD susceptibility of each F2 plant was evaluated on the basis of the mortality rate measured with corresponding F3 progeny seedlings by an in vitro seedling screening method at 4 weeks following inoculation. Analysis of variance was performed for testing the significance of differences between lines and replications (Table 1). The differences between lines were noted to be highly significant and the differences between replications were also noted to be significant. The F2 mortality rate ranged from 0% to 100%, and a frequency of 50–60% class was the highest (Figure 2).

### 2.2. Parental Variety Resequencing

The parental ‘Junam’ and ‘Samgwang’ varieties were resequenced and yielded approximately 55.2 and 31.0 Gbp of raw sequence data, respectively (Table 2). After quality trimming and read mapping onto the Nipponbare reference genome sequence, 322.9 × 10^6^ reads containing 41.7 Gbp, with an average mapping depth of 111.83×, were mapped in case of the ‘Junam’ variety, whereas 200.0 × 10^6^ reads containing 19.4 Gbp, with a mapping depth of 51.94×, were mapped in case of the ‘Samgwang’ variety. Between ‘Junam’ and ‘Samgwang’, 151,916 DNA polymorphisms including 140,318 SNPs and 11,598 InDels were detected. The number and distribution of DNA polymorphisms were highly varied and uneven among the chromosomes (Appendix A). Chromosome 11 harbored the highest number of DNA polymorphisms (*n* = 54,910), while chromosome 8 harbored the lowest number (*n* = 2024). All chromosomes were composed of a mixture of dense and sparse SNP regions (Appendix A).

### 2.3. Genetic Map Construction and Quantitative Trait Locus (QTL) Mapping

We genotyped a total of 188 ‘Junam’/’Samgwang’ F2 plants with 234 KASP markers. Among these, 175 KASP markers yielded reliable genotype data, while the remaining 49 KASP markers yielded unusable data owing to poor allelic discrimination or amplification. Subsequently, a genetic map comprising 175 KASP markers was constructed (Appendix A).

Using the constructed genetic map and data on BD-induced mortality, a QTL analysis was performed using composite interval mapping and a major QTL was discovered at 29.2 cM on chromosome 9 (Appendix A). This QTL was named *qFfR9*, as an abbreviated form of “QTL for *F. fujikuroi* resistance.” The QTL interval, at 95% probability, was 28.1–30.1 cM. The *qFfR9* region was flanked by the markers *KJ09_022* and *KJ09_025*, which were separated by 6.66–7.56 Mbp.

On the basis of DNA polymorphisms between ‘Junam’ and ‘Samgwang’ that were revealed by resequencing, we developed eight cleaved amplified polymorphic sequence (CAPS) markers and one derived CAPS (dCAPS) marker on chromosome 9 (Appendix A), and genotyped ‘Junam’/’Samgwang’ F2 plants. Lastly, we constructed a genetic map comprising 184 markers, including 175 KASP markers, eight CAPS markers, and one dCAPS marker. The total length of the genetic map was 1898.8 cM, and an average interval between the markers was 11.24 cM (Figure 3).

The QTL analysis using the revised map indicated that *qFfR9* was located at 30.1 cM, with an logarithm of the odds (LOD) score of 60.3 (Table 3). The additive effect of this QTL was 35.15 and its dominance effect was −1.27, with an *R*^2^ value of 0.59. The LOD threshold was calculated to be 3.5 through a 1000 times permutation with a probability level of 0.05. The closest markers of this QTL were *KJ09_024* and *9FC30*. The QTL interval at 95% probability was 29.9–31.2 cM. The *qFfR9* region was flanked by the markers *9FC14* and *KJ09_025*, which were separated by 7.24–7.56 Mbp. According to the Rice Annotation Project Database (RAP-DB) (http://rapdb.dna.affrc.go.jp/) database annotation, 15 genes were located in this interval (Figure 4a). A comparison of the genome sequence data of ‘Junam’ and ‘Samgwang’ revealed 208 DNA polymorphisms, including 203 SNPs and five InDels in the genic regions of the genes (Appendix A). The *Os09g0298200* gene did not show DNA polymorphism, while six genes showed SNPs in untranslated region (UTR) or intron or synonymous SNPs in protein-coding sequences. It can be expected that the functions of these genes may not be significantly different between the ‘Junam’ and ‘Samgwang’ varieties. A further eight genes showed non-synonymous SNPs, whose positions in the genes are shown in Figure 4b, in protein-coding sequences and these might exert significant effects on gene function. These eight genes can be considered as potential candidate genes for *qFfR9*.

## 3. Discussion

For mapping a QTL encoding for BD resistance, we used a F2:F3 population derived from crossing BD-resistant ‘Junam’ and BD-susceptible ‘Samgwang’ varieties of rice. These varieties are both Korean japonica rice varieties and share the ‘Hwayeong’ variety as their genealogical parent. ‘Junam’ and ‘Samgwang’ are closely related; ‘Junam’ was bred from a cross of ‘Hwayeong’/’Sangju’/’Ilpum’ and ‘Samgwang’ from a cross of ‘Suwon361’/’Hwayeong’. Previously, we developed 400 KASP makers on the basis of SNPs detected among closely related Korean japonica rice varieties through genome resequencing [24]. These markers can be applied for the mapping and identification of useful genes present in Korean japonica rice varieties. Korean japonica rice varieties show affluent phenotypic variations in important agronomic traits such as disease resistance, pre-harvest sprouting degree, plant architecture, grain size/quality, and so on, providing a high possibility for identifying useful genes. Presently, the development of further KASP markers is underway in our laboratory. Here, we successfully used 175 KASP markers to construct a genetic map with populations derived from the closely related ‘Junam’ and ‘Samgwang’ rice varieties. This demonstrated the practical application of the KASP markers developed by our research group in mapping useful genes in Korean japonica rice varieties. We detected 151,916 DNA polymorphisms including 140,318 SNPs and 11,598 InDels. These affluent DNA polymorphisms enabled the development of a higher number of SNP markers in the QTL region for fine-mapping and selection of candidate genes harboring sequence variations between the ‘Junam’ and ‘Samgwang’ varieties. Taken together, the mapping and identification of useful genes in the Korean japonica rice varieties can be performed more efficiently by combining the genotyping population derived after crossing various Korean japonica rice varieties using KASP markers and resequencing parental varieties.

The previously reported QTLs for BD resistance were located on chromosomes 1, 4, and 10 [6,14,15,16,17,18]. Two QTLs, *qB1* and *qB10*, were found in *RM7180-RM486* region on chromosome 1 and *RM1108-RM304* on chromosome 10 with LOD scores of 2.32 and 2.50, respectively, using a japonica/indica double haploid population [14]. Three QTLs, *qBK1.1*, *qBK1.2*, *qBK1.3*, were found in *RM9-RM11282*, *RM10153-RM5336*, *RM10271-RM35* regions on chromosome 1 with LOD scores of 3.86, 12.07, 2.73, respectively, using recombinant inbred lines derived from a cross between a resistant indica rice variety, ‘Pusa 1342’, and a susceptible variety, ‘Pusa Basmati 1121’ [6]. A major QTL, *qBK1*, was found in a 520 kbp region between *RM8144* and *RM11295* on chromosome 1 with an LOD score of 33.21 with 168 near-isogenic rice lines derived from a cross between ‘Shingwang’, a highly resistant indica variety, and ‘Ilpum’, a highly susceptible japonica variety. Also, in a similar region as *qBK1*, a major QTL, *qFfR1*, was mapped in a 22.56–24.10 Mbp region on chromosome 1 with an LOD score of 22.7 using 180 F2:F3 lines derived from a cross between a resistant Korean japonica variety, ‘Nampyeong’, and a susceptible Korean japonica line, ‘DongjinAD’ [16]. Moreover, a major QTL *qBK1^WD^* was located in a 1.59 Mbp interval of chr01_13542347-chr01_15132528 with an LOD score of 8.29 using the recombinant inbred lines (RILs) derived from a cross between ‘Wonseadaesoo’, a resistant japonica variety, and ‘Junam’, a susceptible japonica variety [17]. In addition, a genome-wide association study with 138 japonica rice accessions revealed two genomic regions highly associated with BD resistance, 413 kbp on chromosome 1 (from position 628,091 to 1,040,823) and 595 kbp on chromosome 4 (from position 31,162,467 to 31,757,436). In this study, we identified a major QTL, *qFfR9*, located in the 7.24–7.56 Mbp region on chromosome 9 with a very high LOD score of 60.3 and an *R*^2^ value of 0.59. This is a novel QTL because no QTLs have been found on chromosome 9 before. The *qFfR9* was found from a high-quality Korean japonica BD resistant rice variety, ‘Samgwang’, and this study is the first one to reveal the BD resistance gene of ‘Samgwang’. It can be inferred that ‘Samgwang’ has a BD resistance gene different from those of other BD-resistant varieties analyzed so far. It can be expected that the marker closest to *qFfR9*, *KJ_09024*, may be used as a selection marker for *qFfR9* in future BD-resistant rice variety breeding programs using ‘Samgwang’ as a resistance donor parent.

The *qFfR9* region contained 15 genes, and 208 DNA polymorphisms were present in the genic regions of 14 genes (Appendix A). Notably, eight genes, which were considered to be candidate genes for *qFfR9*, had non-synonymous SNPs in protein-coding sequences. The *Os09g0298332* gene contained an SNP that resulted in a premature stop codon within the second exon and was annotated as a gene encoding for conserved hypothetical protein. Due to its high-effect SNP, this gene could be considered as a strong candidate for *qFfR9*. The *Os09g0298700* gene showed four non-synonymous SNPs in the protein-coding sequences, and could be a strong candidate for *qFfR9* as well. This gene encodes a protein with an RNA recognition motif (RRM) domain. The domain structure of the deduced protein of the *Os0298700* gene is shown in Appendix A. This protein has two RRM domains and a Spen paralogue and orthologue C-terminal (SPOC) domain. One of the four amino acid changes between ‘Junam’ and ‘Samgwang’ exists in the second RRM domain. Interestingly, an *Arabidopsis* RRM domain protein, ENHANCED DOWNY MILDEW 3 (EDM3), reportedly mediates resistance to the downy mildew disease pathogen, *Hyaloperonospora arabidopsis* [25]. EDM3 controls the alternative polyadenylation of RPP7 resistance gene transcripts by suppressing proximal polyadenylation at a transposon insertion site in the first RPP7 intron. In an *edm3-1* mutant which lost resistance to *Hyaloperonospora arabidopsis*, the level of RPP7 full-length coding transcripts was substantially lower and non-coding proximal-polyadenylated transcripts were increased as compared to the wild type Col-5. If the *Os0298700* gene confers resistance to BD via a similar function to that of EDM3, it might control the polyadenylation of another BD-resistance gene at proximal sites of the gene, which would change the ratio of functional full-length coding transcripts to non-coding proximal-polyadenylated transcripts. Further fine-mapping of the *qFfR9* locus and functional analysis of candidate genes using mutants or transgenic plants are required for the exact identification of genes within this locus.

## 4. Materials and Methods

### 4.1. Plant Materials and Bakanae Disease (BD) Bioassay

The ‘Junam’ and ‘Samgwang’ rice varieties were crossed in summer 2016 in a greenhouse at the National Institute of Agricultural Sciences (NIAS). ‘Junam’ and ‘Samgwang’ are Korean japonica varieties that are susceptible and resistant to BD, respectively (Table 4). F1 and F2 plants were grown in a greenhouse and in the NIAS experiment paddy field, respectively. DNA was extracted from a total of 188 F2 plants, and F3 seeds were harvested from each F2 plant.

A BD response bioassay was performed on the basis of a modified in vitro seedling screening method as described in [16]. Briefly, rice seeds were sterilized by immersion in a 2% sodium hypochlorite solution for 30 min followed by washing with sterile distilled water. The sterilized seeds were allowed to germinate in a tissue culture room at 28 °C for 2 days and subsequently inoculated with a spore solution of *F. fujikuroi* strain CF283 at a concentration of 10^5^ spores/mL. The spores were harvested from the fungus grown on potato dextrose agar media for 1 week in an incubator at 28 °C. The seeds were planted on solid Murashige and Skoog (MS) medium contained in Incu Tissue jars (72 × 72 × 100 mm; SPL Life Sciences Co., Ltd., Pocheon, Korea) 1 day after inoculation. Next, the Incu Tissue jars were placed in a tissue culture room at 28 °C under a 16/8 h light/dark cycle. At approximately 4 weeks after inoculation, when the seedlings of susceptible varieties were mostly dead; the live and dead seedlings were counted and the mortality rates were calculated by dividing the number of dead seedlings by the total number of seedlings.

To determine the mortality rate of each F2 plant derived from the ‘Junam’/’Samgwang’ cross, 60 F3 seeds from each F2 plant were inoculated with the spore solution of *F. fujikuroi* strain CF283, similar to the abovementioned method, and planted in three Incu Tissue jars. At 4 weeks following inoculation, most ‘Junam’ seedlings were dead; live and dead seedlings were counted and the mortality rates were calculated.

To inspect changes in ‘Junam’ and ‘Samgwang’ plant heights, 16 Incu Tissue jars containing 160 inoculated seeds (10 inoculated seeds per jar) and another 16 Incu Tissue jars containing 160 control seeds (10 control seeds per jar) were prepared for each parental variety. From 4 DAI to 32 DAI, the seedlings were removed from two Incu Tissue jars at 4 day intervals and plant heights were measured. To calculate the change in mortality rates of ‘Junam’ and ‘Samgwang’ seeds according to BD, three Incu Tissue jars containing 30 inoculated seeds (10 inoculated seeds per jar) and another three Incu Tissue jars containing 30 control seeds (10 control seeds per jar) were prepared for each parental variety. Mortality rate was measured by counting live and dead plants simultaneously as plant heights were measured.

### 4.2. Parental Variety Resequencing and Marker Development

We extracted genomic DNA from seedlings of ‘Junam’ and ‘Samgwang’ using a DNeasy Plant Maxi kit (Qiagen, Hilden, Germany). Using a TruSeq DNA PCR-free kit (Illumina, Inc., San Diego, CA, USA), sequencing libraries were created following the manufacturer’s protocols, and the fragments of the libraries were paired-end sequenced using the HiSeq 2000 Sequencing System (Illumina, Inc.). The raw readouts that were graded as high-quality using Phred quality values of >Q20 were used to analyze genetic variations between the ‘Junam’ and ‘Samgwang’ varieties. The “Q20”value indicates an accuracy of 99% for the base called. The *Oryza sativa* L. cv. Nipponbare sequence (pseudomolecules IRGSP-1.0, http://rapdb.dna.affrc.go.jp/dowmload/irgsp1.html) was used as a reference sequence. The CLC Assembly Cell program (ver. 3.2.2, http://www.clcbio.com) was primarily used for read mapping and SNP detection. The generated reads were mapped on to the Nipponbare reference sequence using the clc-mapper command with the following parameters: alignment mode, local; similarity, 95%; gap cost, 3; deletion cost, 3; mismatch cost, 2; length fraction, 1.0; repeat, ignore. DNA polymorphisms relative to Nipponbare were detected using clc_find_variation command. On the basis of the detected DNA polymorphisms between ‘Junam’ and Nipponbare and between ‘Samgwang’ and Nipponbare, DNA polymorphisms between ‘Junam’ and ‘Samgwang’ were analyzed using Python programs developed in-house. The detected DNA polymorphisms were annotated as genic and intergenic based on positional information from the Rice Annotation Project Database (RAP-DB, https://rapdb.dna.affrc.go.jp/index.html). DNA polymorphisms in genic regions were classified as coding sequences (CDSs), untranslated regions (UTRs), and introns. The SNPs in the coding region were divided into synonymous SNPs and non-synonymous SNPs by amino acid substitutions. The sequencing data of ‘Junam’ and ‘Samgwang’ varieties have been submitted to the Sequence Read Archive (SRA) database of NCBI (https://www.ncbi.nlm.nih.gov/sra) under the accession numbers SRX5560097 and SRX5560098.

### 4.3. Genetic Map Construction and QTL Mapping

A total of 188 F2 plants obtained from the cross between ‘Junam’ and ‘Samgwang’ varieties were genotyped using KASP markers developed from SNPs detected among Korean japonica rice varieties [24]. Using the Nexar system (LGC Douglas Scientific, Alexandria, VA, USA) in the Seed Industry Promotion Center (Gimje, Korea), KASP amplifications and allelic discriminations were performed. In brief, 1.6 μL KASP reaction mixture including an aliquot (0.8 μL) of 2× Master Mix (LGC Genomics, London, UK), 0.02 μL of 72× KASP assay mix (LGC Genomics, London, UK), and 5 ng genomic DNA template was put in each well of a 384-well Array Tape. The thermal cycling profile for KASP amplification was as follows: 15 min at 94 °C, a touchdown phase of 10 cycles at 94 °C for 20 s, and at 61–55 °C (dropping 0.6 °C per cycle) for 60 s, followed by 26 cycles at 94 °C for 20 s, and 55 °C for 60 s. Next, recycling was performed using three cycles of 94 °C for 20 s and 57 °C for 60 s. Recycling was then performed twice, and the fluorescence read was obtained for KASP genotyping following PCR.

After discovering that a major QTL was located on chromosome 9, CAPS and dCAPS markers were developed on chromosome 9 on the basis of SNPs detected by the resequencing of ‘Junam’ and ‘Samgwang’ varieties. Among the detected SNPs, those lying in restriction enzyme sites were extracted, and CAPS markers were designed as follows: 500 bp left flanking sequences and 500 bp right flanking sequences of SNPs lying in restriction enzyme sites were used to design PCR primers using the BatchPrimer3 1.0 primer design program (http://probes.pw.usda.gov/batchprimer3/). The designed CAPS markers were used for testing parental polymorphisms. PCR products amplified using the primers and parental DNA were digested overnight with restriction enzymes at 37 °C, and separated by electrophoresis on 1.2% agarose gels. Polymorphic CAPS markers were used to genotype F2 plants. Using the SNPs not lying within the restriction enzyme sites, but needed to be converted to markers. dCAPS markers were developed using the dCAPS FINDER 2.0 program (http://helix.wustl.edu/dcaps/dcaps.html).

Using genotypic data on F2 plants, a genetic map was constructed using the MapDisto 1.7 program [26]. The Kosambi function was used as the mapping function. On the basis of the constructed genetic map and BD response data, QTL analysis was performed by composite interval mapping using the Windows QTL Cartographer ver. 2.5 program [27]. The LOD threshold was calculated through a 1000 times permutation with a probability level of 0.05.

### 4.4. Candidate Gene Analysis in the Identified QTL Region

The physical interval of the identified major QTL was deduced from the positions of left and right flanking markers of the QTL interval at 95% probability. The physical map of genes located in this physical interval was drawn from the genome browser of the Rice Annotation Project Database, RAP-DB (https://rapdb.dna.affrc.go.jp/index.html). The locations of DNA polymorphisms between ‘Junam’ and ‘Samgwang’ in the candidate genes were calculated based on the positional information of DNA polymorphisms acquired by the annotation procedure described in 4.2 and candidate gene coordinate information in the RAP-DB.

## 5. Conclusions

Rice bakanae disease (BD) has become a serious threat to almost all rice-cultivating regions worldwide, leading to high (3.0–95.4%) losses in yield. Due to the occurrence of fungicide-resistant strains of BD, the development of BD-resistant rice varieties has become critical for disease control. In this study, we discovered a novel major QTL for BD resistance, *qFfR9*, on rice chromosome 9 with an F2:F3 population derived from a cross between a BD-resistant variety, ‘Samgwang’, and a BD-susceptible variety, ‘Junam’. This QTL and its closest marker, *KJ_09024*, can be used in future BD-resistant rice variety breeding programs, and will be helpful in reducing yield loss caused by BD. Moreover, we found eight candidate genes with non-synonymous SNPs in protein-coding sequences in the *qFfR9* region through a comparison of the genome sequencing data of the two parental varieties. This result will be helpful in identifying the gene for *qFfR9*.

## Figures and Tables

**Figure 1 ijms-20-02598-f001:**
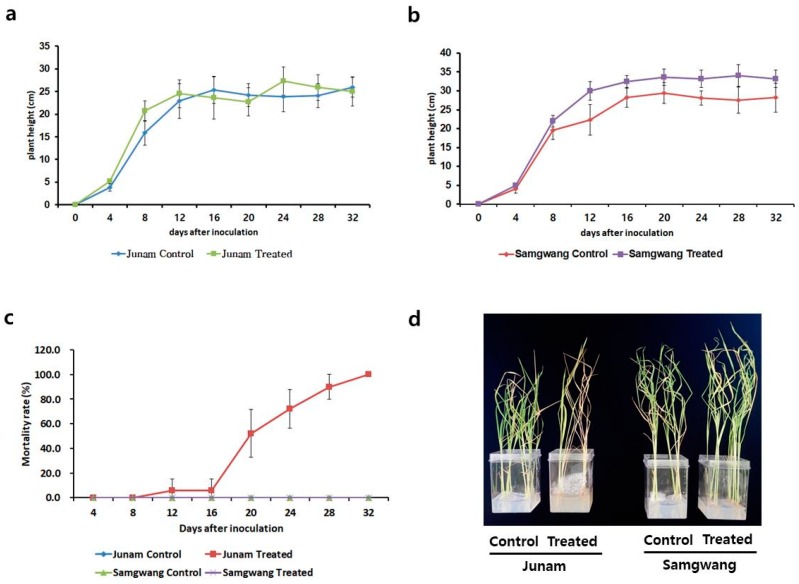
Phenotypic response of the parental varieties to Bakanae disease (BD). (**a**) Changes in plant height of treated and control ‘Junam’ seedlings; (**b**) changes in plant height of treated and control ‘Samgwang’ seedlings; (**c**) changes in the mortality rates of treated and control ‘Junam’ and ‘Samgwang’ plants; (**d**) a photographic record of treated and control ‘Junam’ and ‘Samgwang’ plants at 24 days following inoculation (DAI), indicating noticeable differences in response to BD.

**Figure 2 ijms-20-02598-f002:**
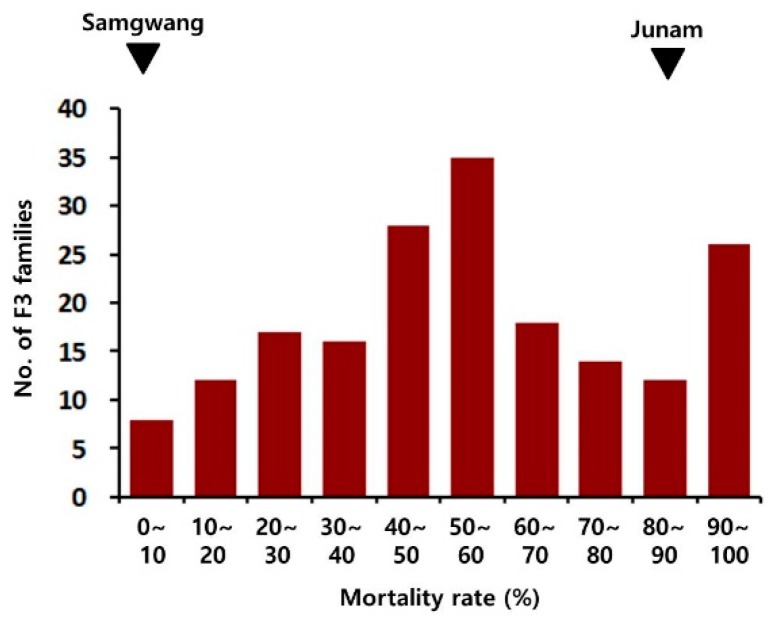
The distribution of the mortality rates of F2 plants obtained by crossing ‘Junam’ and ‘Samgwang’ seedlings. The mortality values for ‘Junam’ and ‘Samgwang’ plants are indicated on the histogram by solid inversed triangles.

**Figure 3 ijms-20-02598-f003:**
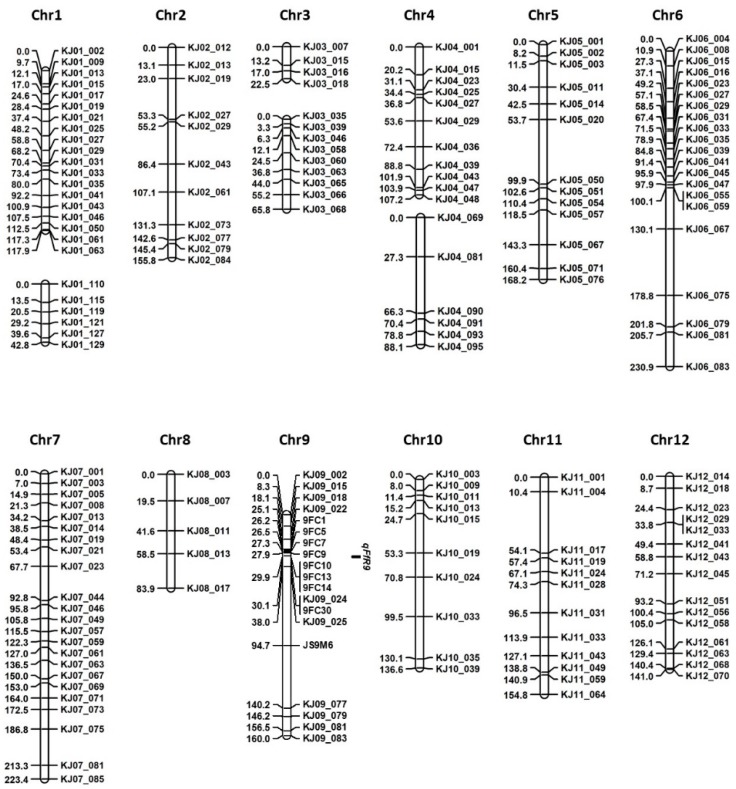
A genetic map constructed using kompetitive allele-specific PCR (KASP), cleaved amplified polymorphic sequence (CAPS), and derived CAPS (dCAPS) markers using F2 plants derived from crossing ‘Junam’ and ‘Samgwang’. The chromosome number is indicated at the top of each chromosome, the name of each marker is indicated at the right side of each chromosome, and the genetic distance of each marker from the first marker at the top of each chromosome is shown on the left side. Genetic distances, measured as centimorgan or cM, were calculated using the Kosambi function. In each chromosome, the linkage groups were separated where the distance between adjacent markers exceeded 50 cM. The quantitative trait locus (QTL) interval at 95% probability of *qFfR9* is indicated by the filled black box.

**Figure 4 ijms-20-02598-f004:**
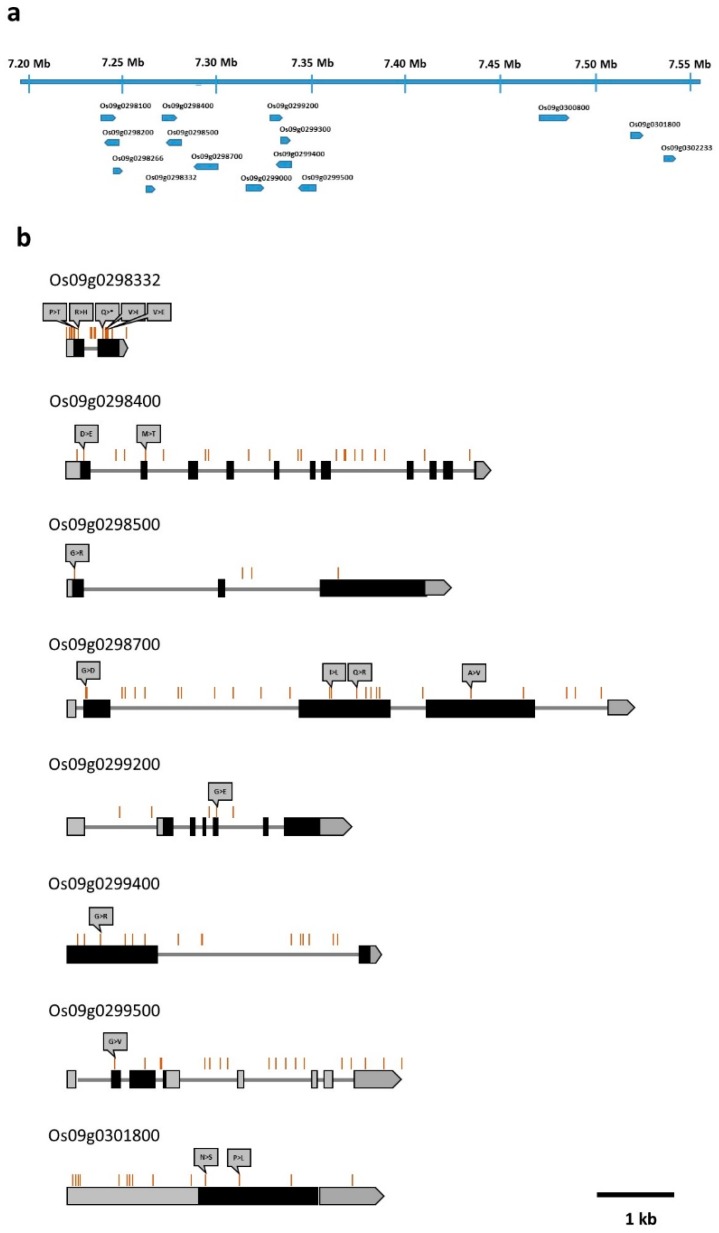
A physical map showing the location of genes in the *qFfR9* region and the distribution of DNA polymorphisms in the candidate genes contained in the *qFfR9* region. (**a**) A physical map showing the location of genes in the *qFfR9* region; (**b**) the distribution of DNA polymorphisms in the candidate genes contained in the *qFfR9* region. The locations of DNA polymorphisms are indicated by brown vertical lines. Filled gray boxes indicate 5’ and 3’ untranslated regions (UTRs), and filled black boxes indicate exons, including protein-coding sequences. Gray lines indicate introns. Changes in amino acids are indicated over non-synonymous single nucleotide polymorphisms (SNPs).

**Table 1 ijms-20-02598-t001:** Analysis of variance (ANOVA) of seedling mortality rate.

Source	Df *	Sum of Squares	Mean Square	*F* Value	Probability
line	185 **	365,459.5	1975.5	16.76	4.1 × 10^−110^
replication	2	863.8	431.9	3.66	0.026545
error	370	43,603.3	117.8		

* degree of freedom; ** Among the 188 F3 families tested, two families were excluded in ANOVA because they have two replication values due to an experimental error.

**Table 2 ijms-20-02598-t002:** Summary of genome sequencing data amount.

Variety	Raw Sequencing Data		After Quality Trimming (Q20 *)			After Read Mapping		
	No. of Reads (×10^6^)	Nucleotides (Gbp **)	No. of Reads (×10^6^)	Nucleotides (Gbp)	Sequencing Depth (×)	No. of Reads (×10^6^)	Nucleotides (Gbp)	Average Mapping Depth (×)
‘Junam’	424.65	55.20	385.75	49.67	133.09	322.89	41.74	111.83
‘Samgwang’	31.18	30.98	241.59	23.39	62.68	199.99	19.38	51.94

* The “Q20”value indicates an accuracy of 99% for the base called; ** “Gbp” indicates 10^9^ bp.

**Table 3 ijms-20-02598-t003:** Identification of quantitative trait locus (QTLs) for Bakanae disease (BD) resistance.

QTL Name	Chromosome	Location (cm)	Closest Marker	QTL Interval * (cM)	LOD **	Additive Effect	Dominance Effect	*R* ^2^
*qFfR9*	9	30.1	*KJ09_024*, *9FC30*	29.9–31.2	60.3	35.15	−1.27	0.59

* Interval at 95% probability; ****** logarithm of the odds.

**Table 4 ijms-20-02598-t004:** Parental varieties used in this study.

Variety	Ecotype	Origin	Response to Bakanae Disease	Parental Cross Combination in Genealogy
‘Junam’	Temperate japonica	Korea	Susceptible	‘Hwayeong’/‘Sangju’/‘Ilpum’
‘Samgwang’	Temperate japonica	Korea	Resistant	‘Suwon361’/‘Hwayeong’

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
