# Peer review of "Rice Genome Resequencing Reveals a Major Quantitative Trait Locus for Resistance to Bakanae Disease Caused by Fusarium fujikuroi"

_ijms, 2019, doi:10.3390/ijms20102598_

Round 1

Reviewer 1 Report

In the Title authors can indicate the scientific name of the disease, Fusarim fukimori.

Around the whole manuscript name of asdsayed cultivars must be between quotations.

Objectives of the work should be separated from the state of the art  eliminating references.

In the Result section 2.1 authors must incorpoarte the phenotypic evaluation of all the assayed material including parental varieties and progenies (Table 2 and Figure 2).

In Figure 3 authors must indicate the identified QTL showed in Table 3.

Figures 4 and 5 should be merged in a unique figure.

Discussion of results must be completed. This is the weaker part of the work. In addition, a conclusión parragraph must be incorporated indicating the main implications of these results for rice breeding and production.

Plant material should be clarified probably incorporating a new table.

A separated section should be included in Material and Methods section clarifing the development and analysis of the physycal map. 

Author Response

Response to Reviewer 1 Comments

Point 1: In the Title authors can indicate the scientific name of the disease, Fusarium fujikuroi.

Response 1: We added the scientific name of the pathogen of bakanae disease, Fusarium fujikuroi, to the title.

Point 2: Around the whole manuscript name of assayed cultivars must be between quotations.

Response 2: We added quotations to the assayed parental varieties, Junam and Samgwang around the whole manuscript except for tables.

Point 3: Objectives of the work should be separated from the state of the art eliminating references. ?

Response 3: We wrote the objective of this work in the last paragraph of Introduction.

Point 4: In the Result section 2.1 authors must incorporate the phenotypic evaluation of all the assayed material including parental varieties and progenies (Table 2 and Figure 2).

Response 4: We incorporated the phenotypic evaluation of all the assayed materials including parental varieties and progenies (Table 2 and Figure 2) to the Result section 2.1.

Point 5: In Figure 3 authors must indicate the identified QTL showed in Table 3.

Response 5: We added the location of the identified QTL to Figure 3.

Point 6: Figures 4 and 5 should be merged in a unique figure.

Response 6: We merged Figure 4 and 5 into Figure 4.

Point 7: Discussion of results must be completed. This is the weaker part of the work. In addition, a conclusion paragraph must be incorporated indicating the main implications of these results for rice breeding and production.

Response 7: We enhanced discussion part by adding comparison between the newly identified BD-resistance QTL, qFfR9, and the previously reported QTLs. Also, protein domain structure analysis of the deduced protein of the candidate gene, Os0298700, was added, and its possible molecular function was discussed.
A conclusion paragraph was added.

Point 8: Plant material should be clarified probably incorporating a new table.

Response 8: A table showing detailed information for the parental varieties was added.

Point 9: A separated section should be included in Material and Methods section clarifing the development and analysis of the physycal map.

Response 9: We added a section "4.4. Candidate gene analysis in the identified QTL region" including the development and analysis of the physical map in Material and Methods.

Reviewer 2 Report

This manuscript deals with QTL detection and picking up candidate genes for resistance to bakanae disease. QTL result was very clear and only single QTL was found,  which have not reported on chr. 9. The focus of the paper was well straightforward.

I have only single concern. The authers described in the last sentense "Further fine-mapping of the QTL". It is necessary to analyse the feature of Os09g0298700 using such as mutant or transgenic plant, not to do fine mapping.   

Author Response

Response to Reviewer 2 Comments

Point 1: I have only single concern. The authors described in the last sentence "Further fine-mapping of the QTL". It is necessary to analyse the feature of Os09g0298700 using such as mutant or transgenic plant, not to do fine mapping.

Response 1: Thank you very much for your kind comments. We added the necessity of the functional analysis of candidate genes using mutants or transgenic plants at the last sentence of the discussion.

Reviewer 3 Report

Bakanae disease resistance in rice is of high importance, since it affects one of the most extended cultivated species. In this paper authors identify a novel QTL associated with this disease that could be useful for identifiying the gene that explains the major percentage of variance. Nevertheless, there are some suggestions that need to be considered. First, the populations used in this paper are F2:F3 were traits are not fixed. I would recommend further analysis in more advanced populations such as F6-F8, where phenotypic variation could be assesed in different environments. Also, a major QTL is described, but no information is provided about minor QTLs detected. Second, the genetic map is a very preliminar version and some chromosomes lack the sufficient number of molecular markers, which affects the QTL detection. Moreover, discussion chapter should be improved. For example authors refer to QTL detected in other chromosomes in previous papers which diverge from the one they detect in chromosome 9. This should be discussed in a more detailed way, explaining for example if it is due to the use of a different rice variety when developing the populations. In discussion authors refers to the LOD score and R2 value. Is the R2 value a percentage meassure? Finally, tables should be more explicative, including leyends for each abbreviation. 

Author Response

Response to Reviewer 3 Comments

Point 1: First, the populations used in this paper are F2:F3 were traits are not fixed. I would recommend further analysis in more advanced populations such as F6-F8, where phenotypic variation could be assessed in different environments.

Response 1: I agree to your comments that advanced populations such as F6-F8 are better materials for QTL mapping than F2:F3 population. But, it is possible to identify QTLs by experiments using F2:F3 populations, and there have been many studies on disease resistance QTL mapping with F2:F3 populations.

Point 2: Also, a major QTL is described, but no information is provided about minor QTLs detected.

Response 2: Even though minor QTLs are not detected, we found a very prominent QTL, and I would like to do further studies for cloning of this QTL by making advanced backcross population.

Point 3: Second, the genetic map is a very preliminar version and some chromosomes lack the sufficient number of molecular markers, which affects the QTL detection.

Response 3: Because the both parental varieties belong to Korean japonica group and share the Hwayeong variety as their geneological parent, they are very close to each other genetically. There should be almost identical genomic regions in genome.  As shown in Figure S1, There are many genomic regions lacking DNA polymorphisms between Junam and Samgwang. The regions lack markers matched largely with those regions between parental varieties. Therefore, it would be not easy to develop markers in these regions.

Point 4: Moreover, discussion chapter should be improved. For example, authors refer to QTL detected in other chromosomes in previous papers which diverge from the one they detect in chromosome 9. This should be discussed in a more detailed way, explaining for example if it is due to the use of a different rice variety when developing the populations.

Response 4: We added comparison between the newly identified BD-resistance QTL, qFfR9, and the previously reported QTLs. Inference on the reason for the difference in QTL location was written according to your recommendation. Also, protein domain structure analysis of the deduced protein of the candidate gene, Os0298700, was added, and its possible molecular function was discussed.

Point 5: In discussion authors refers to the LOD score and R2 value. Is the R2 value a percentage measure?

Response 5: According to the Frequently Asked Questions on QTL Cartographer software, R2 is the proportion of variance explained by a QTL at a test site with the estimated parameters.

Point 6: Finally, tables should be more explicative, including leyends for each abbreviation.

Response 6: We added some more legends to Table 2 and Table S1.

Round 2

Reviewer 1 Report

Authors have revised correctly the manuscript

Author Response

Response to Reviewer 1 Comments

Point 1: Authors have revised correctly the manuscript.

Response 1: Thank you so much for your comments. Thnks to your precious comments, we improved our manuscript greatly.

Reviewer 3 Report

Authors have considerably improved the writing of this paper, and all my previous comments have been taken into account. Even though authors use F2:F3 progenies and more advanced populations such as F6:F8 would have been more suitable, the finding of a major QTL supported by the cloning of the gene make the election of F2:F3 population acceptable. Also, phrase 64 contains a mistake: makor should be changed by major.  

Author Response

Response to Reviewer 3 Comments

Point 1: Authors have considerably improved the writing of this paper, and all my previous comments have been taken into account. Even though authors use F2:F3 progenies and more advanced populations such as F6:F8 would have been more suitable, the finding of a major QTL supported by the cloning of the gene make the election of F2:F3 population acceptable. Also, phrase 64 contains a mistake: makor should be changed by major.  

Response 1: Thank you so much for your comments. Thanks to your precious comments, we improved our manuscript greatly. We corected the mistake by changing the word 'makor' to 'major'.